# The Role of Spirituality during Suicide Bereavement: A Qualitative Study

**DOI:** 10.3390/ijerph19148740

**Published:** 2022-07-18

**Authors:** Austėja Agnietė Čepulienė, Paulius Skruibis

**Affiliations:** Suicidology Research Center, Psychology Institute, Faculty of Philosophy, Vilnius University, 01513 Vilnius, Lithuania

**Keywords:** postvention, suicide bereavement, spirituality, religion, religious coping, spiritual coping

## Abstract

Background: A loved one’s loss due to suicide can be a traumatic experience and trigger a complex grief process. Although spirituality, defined as a search for the Sacred in a broad sense, can be a resource and an obstacle in coping with the suicide bereavement process, there is a gap in scientific understanding of the role spirituality plays during the process. Methods: To explore the role of spirituality in people bereaved by suicide, we recruited 11 women who lost a life partner due to suicide. We conducted semi-structured interviews and analyzed the data using reflexive thematic analysis. Results: We identified the following three themes: spirituality is a supportive resource that can be reached for or achieved without conscious involvement; spirituality provides helpful ways to cope with grief; spirituality makes the grief process more difficult. Conclusions: Spirituality, if personally meaningful and supported by others, can function as a resource after a loved one’s suicide and even add to post-traumatic growth after the loss. On the contrary, spirituality-related issues, such as stigmatization and a lack of personally meaningful traditions, can distress the bereaved. Difficult spiritual experiences and questions can become an intricate part of the grief process.

## 1. Introduction

Every year, around 700,000 people die due to suicide [1]. In Lithuania, where the current study was conducted, the suicide rate is around 26.1 per 100,000 [1], which is more than double the average. Each suicide leaves approximately ten times more people bereaved by suicide [2]. Such loss is a painful experience that can negatively influence different aspects of life, such as psychological [3] and physical health [4] and relationships [5]. As another potentially important dimension of life, spirituality can also be affected by loss due to suicide [6,7]. However, spirituality-related experiences and beliefs, if not too shattered by the loss, might also function as a resource during suicide bereavement [6,8].

Spirituality is a complex phenomenon which is difficult to define accurately. The ongoing discussion of spirituality’s relationship with religiosity and religion makes the definition even more complicated. Earlier viewed as a deep religiosity [9], today spirituality can be seen as not connected to religiosity [10,11] or connected to religiosity by being a part of it [12,13]. Spirituality, separated from religiosity, can be defined as a personal need, experience, and by ways to connect to transcendence, e.g., [14], including concepts of Higher Power and supernatural or not including these [14]. Personal spirituality can be experienced and practised in various ways, such as through music [13,15], relationships or nature [13]. Religiosity can be seen as an affiliation to a religious organization consisting of internalized beliefs and values, performing religious rituals and practices [16]. The spiritual part of religiosity would be a personal inner connection to the Higher power [13,17]. Some authors suggest that because of the broadness of the definitions of spirituality, it is essential to what spirituality is for a particular group of people or to choose one of the definitions for the specific research [18].

In the current study, we chose to see spirituality as a search for the domain of the Sacred [13]. To delve deeper into understanding the phenomenon, we also made use of analytical psychology’s ideas. Analytical psychology, created by C. G. Jung, offers extended explanations of psychology, religion and spirituality. According to analytical psychology, spirituality is a human need to connect with something bigger, a subjective and live experience [11]. It can differ from connecting to God or Higher power, to nature or virtues [13,17]. In a way, spirituality can be seen as a process of connecting to the archetype of the Self—a hidden guiding part of the psyche, which gives the potential to connect consciousness with unconsciousness [19], which can be defined as part of the psyche, which contains personal material that has been repressed or simply forgotten and the collective material of the inherited experiences of all humankind that guides the conscious mind [19]. According to this definition, religiosity in the current study was seen as a method of searching for the Sacred, therefore, as a part of spirituality chosen by a person [11,13]. 

A loved one’s death, in general, but also due to suicide, can evoke spiritual questions. Therefore, facing the death of a loved one can be a turning point for the search for the Sacred [6,7,20]. The loved one’s death reminds the bereaved of the temporality of life, which raises questions about the purpose of life, beliefs in the afterlife, or the existence of the soul [6,7,8]. Spiritual experiences, which can strike the bereaved person, evoke questions about the nature of those experiences [8,21,22,23]. If a person is highly religious, the answers might be easier to find [24]. However, the questions and experiences also visit non-religious people [8], which adds the interdisciplinary phenomenon of spirituality to the psychological issues of suicide bereavement.

When the cause of death is suicide, the search for the Sacred can become even more complicated. Since suicide is not a religiously appropriate death [6,25], the bereavement process can be influenced by stigmatizing or self-stigmatizing attitudes [6,7,25]. Suicide is hard to comprehend and leaves a person with a haunting question of why [26,27], which can cause the unsatisfied need for spiritual explanations, why the tragedy happened, or where God was when the loved one died [6,8]. The feelings of disappointment and distrust in Higher powers can cause a spiritual crisis [6,7,8] and the need to re-estimate one’s beliefs and virtues. All those experiences related to spiritual issues add to the challenging grief process. 

Despite the complications related to spirituality during grief, some aspects of spirituality seem helpful during suicide bereavement. Spirituality and religiosity can be potential resources during bereavement after a loved one’s death [21,28,29]. The religious community can support the bereaved [24,25] if it withstands stigmatizing attitudes and behaviors [8]. The issues related to belief in the afterlife can be eased if a person has higher religiosity [30] or if the bereaved person perceives spiritual experiences as proof of the hereafter [21,22]. Beliefs in the afterlife can satisfy the need to continue the bond with the deceased as a further existing figure [8]. 

The positive and negative religious coping models might be applied in the context of suicide bereavement [31]. Positive coping refers to using religious and spiritual methods, such as religious forgiveness, seeking spiritual support, and coping in a community, which can help with better health and adjustment outcomes after difficult life events [31,32]. Negative religious coping refers to discontent in religious and spiritual matters or, for example, feeling as if one is being punished by God, which has negative correlations with adjustment after difficult life events [31,32]. Spirituality and positive spiritual coping could function as a resource during suicide bereavement. Nevertheless, it depends on at least several specific aspects of how the person understands spirituality if they belong to a religious community or what their beliefs in the afterlife are.

A few scientific studies and reviews about suicide bereavement and spirituality indicate the topic’s complexity and importance. The research about spirituality during suicide bereavement is limited by several quantitative studies, which reveal that the bereaved by suicide often have spiritual experiences, which are helpful with adjustment [22]. Belief in the afterlife can help with adjustment to bereavement if the bereaved has the hope to meet the deceased again [30]; the bereaved tend to seek spiritual support and advice [22,33]; and 61 percent of parents bereaved by a child’s suicide struggle to find meaning in the loss [34]. Qualitative studies reveal that a loved one’s loss due to suicide can strengthen spirituality regardless of the person’s previous beliefs [6]. Religiosity can help answer questions regarding the afterlife [25] and solve questions about responsibility [35]. Spiritual experiences are essential for transforming the continuing bond with the deceased [6,23]. Spiritual rituals help to sooth oneself after the loss [7]. Religious social support is vital in practical, emotional, and spiritual ways [25]. A loved one’s suicide can also induce a spiritual crisis [6,7,25] and perceived or real stigmatization from the religious community [6,25,35]. The mentioned studies focused only partially on spirituality, religiosity or suicide bereavement processes. They indicated the need to conduct a focused qualitative study on the broad phenomenon of spirituality during suicide bereavement [6,7,8,25]. A better scientific understanding of spirituality and suicide bereavement could help reach a more profound knowledge about suicide bereavement in general and formulate guidelines for practitioners who help people who suffer after a loved one’s suicide. Therefore, the current study aimed to qualitatively explore the role of spirituality during suicide bereavement. 

To explore the phenomenon, we decided to ensure the homogeneity of the research participant’s group [36] by conducting interviews only with women who lost their life partners (husbands, fiancées or romantic partners with whom they lived before the loss). The group of women were chosen because, in Lithuania, where the study was conducted, there is a huge gap between male (36 per 100,000) and female (6 per 100,000) suicide rates [1]. Masculinity is also considered a risk factor for suicide in different countries [37]. Since more than 90 percent of people see themselves as heterosexual [38], the people who suffer a life partner’s suicide are primarily women. The suicide bereaved life partners can have even more expressed complicated grief than friends, parents, or colleagues [39] and children or siblings [40]. This can be explained by challenges related to financial or housing issues, difficulties of suddenly becoming a single parent, and loneliness related to the loss of a partner such as the loss of the intimacy and ability to share the feeling of grief and feelings [40]. Additionally, people bereaved by a partner’s suicide can experience more anger toward the deceased than parents would feel for a deceased child [41]. Nevertheless, partners receive less emotional support than those bereaved by other causes of death [42,43]. The partner’s suicide increases the bereaved’s suicide risk [44] and worsens general mental and psychological health [4]. Therefore, the wellbeing of this specific group of bereaved people is an essential focus for scientific studies of experiences and help strategies [45]. 

## 2. Materials and Methods

### 2.1. Materials

For data collection, we chose to conduct semi-structured face-to-face interviews consisting of the following 3 main questions: “Tell me, please, about your loss”; “What is the role of spirituality during your suicide bereavement?”; “How did spirituality-related topics reveal themselves concerning relationship to others (religious community, other people, the deceased)?”. After the interviews, we asked the participants about their primary demographic information.

### 2.2. Participants

We interviewed 11 women aged 28 to 62 (see Table 1 for more details). Ten interviewed women were Lithuanian, and one was Lithuanian Russian. Nine lost their husbands, one her fiancé and one her romantic partner, due to suicide. Time elapsed since suicide ranged from 2 to 5 years. The women were in a relationship with the deceased men from 1.5 to 39 years (17.7 years on average). All of them lived with the deceased men at their death. Eight research participants had one or more children with the deceased. Nine women had higher education and two professional education. Eight women at the time of the interview lived in a city, two in a town, and one in a rural area. Five of the participants ascribed themselves to Catholicism, one to the Eastern Orthodox Church, one described her affiliation as “Catholic, but with a question mark”, one participant described herself as Catholic but not practicing the faith, two participants had no religious affiliation, and one described herself as having no affiliation or belonging to all religions. Nine of the deceased men died by hanging, one by overdose and hanging, and one by cutting his veins. The average age of the deceased was 41, ranging from 30 to 57.

### 2.3. Procedure

We used purposeful and snowball sampling to reach participants who fit our research aims. We spread the invitation through emails and social media posts to members of the Lithuanian psychologist alliance, practicing psychologists, social media followers, and friends to reach research participants. We also invited potential participants through educational articles in online and printed newspapers and an educational talk on the radio. Lastly, we searched for participants proactively, calling on the participants (who earlier consented to be invited for other studies) of another research project conducted in the suicide research center. We asked the participants to contact the first author through email or phone by themselves or to fill out a short questionnaire and be contacted by the researcher later. Eight people filled out the questionnaire, and eight of them participated. We called fourteen people proactively, and five agreed to participate (see Table 1).

We collected the interviews between January 2021 and November 2021. Because of the COVID-19 situation and nationwide lock-down, we provided the participants with a choice to provide the interview through a video call with the program “Zoom” or in person. Although we were aware that in-person and video call interviews might differ because of differences in creating contact with the participant and the participant’s sense of privacy, studies suggest that the differences might not be significant for the duration of the interview, the number of codes, and the interview analysis [46,47] or sharing deeply personal experiences [48]. We conducted 7 interviews through video calls and 4 in person in the first author’s consulting room. The interviews lasted 1–2.5 h (see Table 1). The first author carried out eight interviews. The other three interviews were carried out by master’s degree students who were members of the project and were trained by the first author to reach consistency.

Before the interview, all participants provided written or scanned informed consent. Each participant received an identification code, so they could withdraw from the study at any time (even after the interview) and maintain anonymity. After the interviews, we provided the participants with written information about emotional and psychological help resources and contact details of researchers in case they needed to talk about their feelings and thoughts after the interviews.

### 2.4. Data Analysis

We audio-recorded, transcribed and coded each interview. To reflect on the potential researchers’ influence on the data analysis, from the beginning of collecting interviews until the final themes were created, the researchers who analyzed the data kept a research diary in which they reflected on their feelings, thoughts, insights, memories, and interpretations of the data. The personal involvement of the first author was exhaustively described in a separate article [49].

We chose the phenomenological approach to analyze the data, which explains that the lived experience of the research participants allows the researchers to explore the chosen phenomenon from their subjective reality. The approach insists that, as limited human beings, we cannot capture the whole of objective reality (if it exists). We can only come as close as possible to the subjective realities of other people. Qualitative methods, such as thematic analyses, combine these subjective realities and find meaningful patterns in the experiences [50]. We used a reflexive thematic analysis [51,52,53,54] to identify the main themes in the data. We chose this method and an inductive data-driven analysis of the data [51] because they were deemed a good match for the current study question, which is oriented to the previously not researched exploration of experiences related to spirituality during suicide bereavement.

We conducted a reflexive thematic analysis following the steps provided by Braun and Clarke [51,52,53,54]. The coders listened to the interviews and read the transcripts to familiarize themselves with the data. For initial coding, we used Atlas.ti Web [55]. The first author coded six interviews. Codes were reviewed by master’s and bachelor’s degree students who were members of the current research project. Members of the research project coded five other interviews, and the first author reviewed the codes. The first author then organized the codes to the potential themes and sub-themes. The second author reviewed the initial thematic map, and the disagreements were solved by discussion. Afterwards, the final thematic map was created.

### 2.5. Ethical Approval

The Psychology Research Ethics Committee of Vilnius University approved the study (22 January 2021, Number 56).

## 3. Results

From the data, we identified the following three main themes followed by subthemes (see Figure 1): spirituality is a supportive resource that can be reached for or achieved without conscious involvement; spirituality provides helpful ways to cope with grief; spirituality makes the grief process more difficult. We discuss the themes below.

### 3.1. Spirituality Is a Supportive Resource That Can Be Reached for or Come without Conscious Involvement

All the participants spoke about spirituality as a supportive resource. The resource sometimes was activated through active involvement, while at other times it functioned without one’s conscious efforts.

#### 3.1.1. Changed Relationship with Spirituality after the Loss

Eight participants noticed a changed relationship with spirituality after the loved one’s suicide.

Participants felt new abilities to connect with a Higher power after the loss:


*My service is like this, prayer service, during glorification, when I have to pray with my community, and ask the spirit, what the spirit says not just to me personally, but to the community, then the prayers too are strong and they just come and you don’t need to translate the scripture, words from the scripture just come to consciousness and then I say them, translate them to people who are near me and for me… I make sure that God is talking to me, I don’t remember anything by heart, but I just read, and God gives me strength to remember exact things that he wants me to say.*
(Agnė, 36)

Several noticed the intuitive understanding that the loved one had died and where to find the body:


*I don’t know, I just turned, and some force pushed me into the child’s room. And I go into the room, turn to the left and he was just there on the wardrobe.*
(Asta, 36)

Many participants noticed that their spirituality strengthened after the loss. This manifested through a clearer understanding and holding onto one’s virtues:


*To have those base virtues and comply with them. Like I said, I never blamed anyone for this situation (for husband’s death), didn’t blame. hh the one everyone attacked (blamed for the husband’s suicide).*
(Elena, 34)

Participants also felt strengthened trust in people and relationships:


*But like I said, we concentrate on the negative things and can’t see anything else. And here is where Lithuanians seem genuine. Some of course I met horribly. I am also saying this as a Russian and it doesn’t depend on nationality, it’s just about the person… So now I understand that all of us, we are all the same, absolutely the same. And I felt amazing, even if I say that there are humanely humans, that it still exists.*
(Asta, 36)

Several participants talked about strengthened self-worth and emotional endurance after the loss:


*But you know, when a year passed, then I felt that I am strong. And I stopped getting mad about their comments. I felt that no matter how hard it is, I manage the kids well.*
(Rasa, 41)

Some noticed the ability to accept the relative insignificance of temporal human life:


*When you start to become aware, create a better world, to do something, then it doesn’t have any meaning. And somehow like I said that understanding, it came, and there is no point to it, you just must experience it and spirituality is part of the experience—to not harm another, not harm nature, to create, to leave something better when your gone, it doesn’t have to be visible, and people forget really fast, really fast. It’s an existential, like you’re nothing.*
(Nida, 35)

Others spoke of strengthened religiosity after the loss:


*I feel like a giant comparing my spirituality five years back. Because there is a different quality, a completely different level of faith, communication with God and the work with community, worship. It’s like God says, I will give you an abundant life, and now I have that abundance. Abundance, the fulfillment, in any way, I have everything.*
(Agnė, 36)

Some of the participants felt support from a Higher power after the loss:


*And throughout the mourning, during the funeral, I felt as if I was being carried on arms, I don’t know how to say it, I actually felt, that I was being carried by God, and I couldn’t by myself, I saw his mom, and his mom was hysterical, going crazy, and she suffered terribly, and there was a lot of guilt there, childhood did its own thing, i understand, that feeling of guilt, but I also saw what kind of pain one can feel if they don’t believe in God, when the person is alone in that pain, he is alone and he doesn’t have anyone to call upon, he doesn’t have anyone to rely on, like it says in the psalms, if thousands fall from the right and thousands from the left, you will be untouched and I felt like the world had fallen, it was falling, and I was left with two small kids, I didn’t have a job, but I felt, that I am in God’s hands, in his embrace and I never felt guilt, I was just sad, really sad, and I understood his pain.*
(Agnė, 36)

#### 3.1.2. Believing in a Higher Power Is Helpful with the Meaning-Making of the Suicide

Believing in a Higher power and its function in a person’s life helped participants make meaning of the suicide. Six participants spoke of thinking that the Higher power was preparing them for the suicide of the loved one, which would be proof that the suicide was in some ways supervised by the Higher power:


*And another thing that supported me, well supported, I unconsciously knew that someday it will happen. Even though you don’t think about it. Because like I said, since I’m interest in esotericism, I was in regression some time ago. In a regression seminar. That you can go back, go to the future. It’s a conscious dream. And that was seventeen years ago. A long time ago. And then I dreamed about a different country, where my future husband works, town, all their buildings, I remember how I explored it. I dreamed about myself with kids in mourning clothes. I was wearing a black dress and the three of us with kids, the husband wasn’t with us. And then I cried, o God, how I cried. I’ve never cried as much as I did then.*
(Jurga, 49)

Participants also used ideas related to spirituality to understand the reasons for suicide. This included religious explanations about human suffering, the importance of accepting the decisions of others, trust in God’s will or believing in fate. For example:


*There was suicide in his family, his grandfather ended up under the tracks, or something, it looks like that, whoever believes in those things. I think it’s of the same matter and maybe he finished it in his family.*
(Laima, 28)

One participant believed that the deceased was obsessed with demonic powers:


*I absolutely saw I mean that we have a situation and now looking back, we had a situation with the ghost of the dead and when the exorcist prayed for him and asked him, where do you feel in the body, some sort of sensation right, they make conclusion according to that, diagnosis and he said in the spine.*
(Agnė, 36)

Some participants viewed the loss as a test sent by God. The bereaved felt the need to prove to God the integrity of their faith:


*It looks like maybe he (God) wanted to make me stronger and that I would be ready for my current life.*
(Asta, 36)

Some of the participants believed that the deceased was preparing himself spiritually for the suicide, which helped with the relief that the loved one is happier in the afterlife:


*In the end he talked more about God and believed in him, and he talked about the last judgement day or something, about a refinery. That’s why he also encouraged me and somehow, he was slowly taking control, I mean, to not do anything bad. Because I thought maybe that he was sorry or not sorry, but I just think: “well yeah, however much it hurts me, but he doesn’t have any problems there (in the afterlife).”*
(Rasa, 41)

#### 3.1.3. Experiencing Compassion and Responsibility for Other People during Grief

Seven participants spoke of the strengthened compassion and responsibility for other people. Participants revealed that experiencing compassion and responsibility for their children or other people was a way to find the strength to live on:


*It was really difficult at first, but it’s like I said two little kids and it’s not okay. You have someone to take them, you can’t just put them aside, you must stand up, stand tall and move forward. So in this case my focus is on the first days.*
(Elena, 34)

The wish to help others who suffered after similar losses helped regain some of the control over one participant’s life:


*The journey continues even now slowly, like the muscles, the spiritual ones, they are stronger now, the first step shows this, that I want to do that group (self-help group) so I believe that this is a sign that if I can help others, that it’s a sort of healing process. If that energy, when I start to move and I’m ready to share with others, the balance is being restored.*
(Liepa, 49)

The wish to participate in the current study was also perceived to help others who suffer:


*Right after the loss, I received one questionnaire, then another, and my mom said stop, let it go, why do you need this, to reopen the wound, she said, don’t you understand, I rethink the event over and over and over again, and a thousand times more, and I said it’s not getting any better, and… if I’m doing a good job by sharing my thoughts, if they are worthy or not, but I say, maybe it’ll help someone, maybe someone will see that moment, when it happens.*
(Dalia, 62)

#### 3.1.4. Connecting to Spirituality during Grief

Five participants spoke of the need and active ways by which to connect to spirituality during grief. For example, the ways to connect with spirituality during grief involved active praying:


*I never knew how to pray, because it always seemed, that there are some kind of rules, how you should address… If it’s God, or the earth, or what you believe and somehow I thought I’m not doing it right and then I asked my grandmother, so how should I pray now, in your own words, how you want to, that’s how you should pray, you’ll be heard anyway, or so I would say a worldly prayer, the one for the dead of course, because I thought, that it is necessary. And then in my own words I’d just pray, I actually would just ask, not pray, that I wouldn’t go crazy, that I’d be given strength, to be here, but I prayed to him. Yeah. For an easy path, I prayed for his forgiveness, somehow my prayer was related to his easy departure.*
(Laima, 28)

As well as spending time in a sacral place:


*And that is why, why I say that it would happened in church. I would want to kneel, to lean on those wooden things, to lay my head and cry. And somehow that atmosphere would provoke that reaction, those organs, when they play. And the priests’ sermons, it always seemed that it was talking about me, or about him, or about our family. Somehow everything seems to be happening at the right time and place.*
(Rasa, 41)

Or listening to music, which started to have spiritual meaning after the loss:


*It has a special effect on me even now, mass and the sacral music, singing, it’s unbelievable, maybe it’s even painful, the music. I always, it tears me up.*
(Laima, 28)

#### 3.1.5. Experiencing Continuing Bond with the Deceased without Conscious Decisions

Six participants experienced continuing bonds with the deceased, which they comprehend as occurring without a conscious choice being made. They felt that the dead initiate communication and help the bereaved:


*Later I was going to work, a few years after his death, I was sleeping, and with his knee to my side boom boom near the bed, get up, boom boom, get up, I look aaaa it’s 8, I overslept, that means before I opened the store, at that moment and my subconscious connected my alarm, that I need to go to sleep, but I thought that I felt him, that he is waking me up, that honey, wake up, go to work, was him, and I don’t know how to say it, I just felt that substance. It seems that the air is denser and if you turned on the light you’d see that it’s him in layers, like turning the light on in a fog, this way and that way, and when you catch this fog with a flashlight, from one corner it is clear, but from another you see the layers, it’s an impression, or it’s my experience talking, if it’s real, I want to believe it, that it can actually happen.*
(Dalia, 62)

The communication from the deceased mainly was perceived as comforting and provided relief:


*And I dreamed that he… As if there was a party and he came a bit drunk, but he was in a good mood, happy. The only thing is that he didn’t come near me. His brother was there. And he went up to his brother to talk and I hear what he is speaking. And he was interacting with him in a brotherly way, in a manly way. And he tells him: “you know why I love my life? Because, however much money I spend, there will always be more. Whenever I come back home, nobody is mad at me and (starts crying) whatever sins I commit, He always forgives me. And “he” as I understand, was Jesus, I knew that very clearly for some reason, that he was in His care, in that other life. And there is eternal joy, and he doesn’t have any problems there… It affected me deeply, it was very strong, because… You know some dreams are very vivid, others not so much… but that dream, it’s like I said, it seems I see it clearly, word for word. And somehow, I felt that … you know this was after, wait after, after five months, or four months after his death, that type of dream. So I don’t know, it seems to me, that was the first, clearly some connection from him.*
(Rasa, 41)

To provide an overview, spirituality and spiritual experiences during suicide bereavement can function as a supportive resource, which helps to explain why the suicide happened and provides spiritual strength to cope with grief. Spirituality can be experienced as coming from outside without consciously thinking about it or asking for it. This involves noticing changed and strengthened spirituality or religiosity, feelings of more profound compassion and responsibility to others, and experiential manifestations of continuing bonds. Active praying or travelling to sacral spaces activated the connection to a Higher power and provided comfort after the loss.

### 3.2. Spirituality Provides Helpful Ways to Cope with Grief

All research participants spoke of how spirituality and religiosity provided ways to cope with grief. We discuss them one by one below.

#### 3.2.1. Rituals to Ease the Difficulty of the Grief

If having personal meaning, for nine participants, rituals seemed to ease the difficulty of the grief. The funeral ritual was perceived as an essential part of the bereavement process. For some, organizing the funeral provided a short break from the loneliness of grief:


*People were lost after a sudden death. They experience it the same, they are in a period of shock. Then they, if they continue the burial service, all those things and have the energy to do all of that. But usually, they don’t want to be alone, they want to take care of everything.*
(Karolina, 46)

For others, funeral and fare-well rituals were a way to express the last respects for the deceased:


*But I said right away (giggles)… find me the best hall, the best place. I will pay whatever you want, but I want the best… There were no questions because he was also always the one to choose the best. Or choosing closest to the best. Money was spent so that it’d be nice. I knew that people would come, and you need to, so that the funeral would be, how to say so that he would truly be respected.*
(Jurga, 49)

The ritual of a funeral also helped to comprehend that the loved one died:


*What I understood, when his casket was being buried, I actually understood that he is not here, that this is a body, I thought of this often.*
(Agnė, 36)

Participants spoke of the many different forms personal rituals can take such as travelling and being in nature:


*So those journeys and world knowledge and new knowledge of nature, that you climb the mountain, it means those special paths right, and you get to the top and there is this huge wind, and you can’t think of anything else and it’s a new place for you and a place you can’t find in Lithuania, you start to think about the world differently, new colors appear. You understand that… there is more of everything and it’s different from what you’re used to. Or when you stand up, on salt, the lake of salt right where it is white and there is no horizon, and you are in the middle of nowhere. You just hang in the middle and you understand that there is something… Something magical.*
(Nida, 35);

creating something which could be related to emotional expressions:


*My nature is creative, and I am creative, so I searched through drawing and my hobby allowed me to find that, what I did in my childhood, I drew a lot, was picky, sewed and still I came back to that after the funeral, I found wool, I found wool and looked at the process in a more creative way.*
(Liepa, 49)

meditating and doing yoga:


*In that hour and a half when you do yoga, you just, translate stress, anxiety, tension, thoughts through the body, because when you do it, you don’t think about other things, you just think how to do yoga. How to bend your legs, how to bend your body and other stuff. And in the end, there is shavasana, where you just lay ten minutes, it’s called the dead pose. You must disconnect and not think of anything. So, this is the best thing, when you lay down, open your body, and not see or hear anything anymore. That is equal to meditation actually. And yeah. This thing was taught by the teacher and told us that when you come to yoga it’s like a new page, you forget everything that is around you, and start from scratch. Actually, that helps a lot.*
(Jurga, 49)

Even self-analyzing was considered a spiritual experienced in a psychologist’s room or a helpful evening ritual:


*Just after my husband’s death I had space, usually during my talks with my psychologist. And you know that once a week you will complete that exercise either way… I tried to and each evening, right before going to sleep to think about the day… with completed facts, put into drawers and not worry about things, that are out of your control.*
(Elena, 34)

Several participants perceived gratitude as a meditation or a prayer as an antidote for suffering:


*There are people, I started to move from household stuff to people, God and those people exist, and I have a job and friends and my relatives don’t turn from me and then everything expands, I have so much and to be grateful, and then joy comes from gratitude and then I understood, I concentrate on what I don’t have. And that is a huge problem. And when I started to focus on what I do have, how much there is and health, and that my kids are healthy and that is amazing.*
(Agnė, 36)

Personal rituals related to fire as having a cleansing, easing, and finishing power, were necessary for some of the participants:


*And when we talked about fire, then I just wanted to burn something. Because Rimas (the deceased) wanted to be cremated. So, fire, even to him, seemed to cleanse and that it will somehow show all those bad things and fire is powerful, that it just burns (whispers) just burns.*
(Eglė, 31)

Lastly, religious rituals such as attending mass every week for several participants were found to be helpful after the loss:


*This is about being religious, or how to say it, but I know now how much it snowed, how much it winded… but I walk there… and my knee joint is week, I have a hard time walking, fifty meters is a lot… I must go to church… so I had to walk a kilometer and three hundred meters.*
(Dalia, 62)

#### 3.2.2. Spirituality Related Attitudes Help during Grief

Nine participants spoke of helpful attitudes related to spirituality. This involved beliefs about death not being the end:


*How to say… I’m a believer in energy and energetic bodies. I’m not sorry when someone dies and I usually don’t cry, because I know that around our body… I can feel other bodies and that consciousness is not in the body, but it’s… you can expand your consciousness to eternity.*
(Eglė, 31)

Other spirituality or religiosity related beliefs were also helpful, such as accepting the uncontrollability:


*As for spirituality, in the sense of religion, I discovered specific philosophical points in Buddhism. Through literature, I started to read more books that helped me… I found that the whole theory of mindfulness came from Buddhist philosophy… The simplicity of philosophy and the simplicity of being has helped me restore that spiritual balance… I am suffering because I wanted everything. It is that desire to have a person, the desire to be attached… the desire to control that situation… Well, it does not happen that way. This understanding of mind through mindfulness… and all these fundamental truths and points of reference in Buddhism may have helped restore that spiritual balance little by little… some kind of mantra, as some kind of reminder that no one has promised you anything.*
(Nida, 35)

Similarly, the idea of not being able to control many aspects of life, and trusting in a Higher power or God’s will was helpful:


*What more to hang on to? Some kind of will of God that God knows what he is doing. To trust Him and His plan. Maybe it is awful for me now, it is hard, but apparently, for some reason, it must be that way.*
(Rasa, 41)

#### 3.2.3. Other People Can Influence the Bereaved’s Spirituality after Loss

For eight participants, communicating with other people in a spiritual context provided another spiritual way to cope. Participants spoke firstly about the need for psychological or emotional support and spiritual help after the loss. For some, it was hard to find the service:


*The possibility of getting some information to reach or go on that kind of spiritual path… we do not have such very easily accessible information… like could you help me here now, if not for some kind of psychological help, but let’s say something like that.*
(Karolina, 46)

For some participants, spiritual help seemed more helpful than psychological help:


*Like even psychologists cannot share advice. They must force a person to choose some solution. I received honest, real help (from the clergy) when I was told to thank the Lord. I was angry at first for such advice… However, I tried to practice what he told me, what was being advised to me, and it worked, and it actually worked.*
(Agnė, 36)

Several participants said that psychological help was related to spiritual support. For example, a psychologist helped to understand the meaning of a funeral:


*The cremation itself and saying goodbye before an essential aspect I talked about to my psychologist… That here is the fact of death. Well, that is why… the funerals usually are for the comprehending that it will not be like that anymore, that the husband will not be here anymore.*
(Elena, 34)

Many participants spoke of a very beneficial relationship with a religious person after the loss. Some of them befriended a priest, who helped with reducing guilt after a loved one’s suicide:


*We have been with him so far. We have become like a family. We do not… discuss all those religious issues and so on. He is so humanly, so secular. Without those, you know, those constant sermons… He says: why are you torturing yourself… he (the husband) has chosen such a path, an easy path because it is much harder to fix problems than to die… he left you with the child, with your problems… You would be guilty if, say, you would hold him by the hands… and put his head in a loop, tighten it and push the chair… to the side, then you might be guilty of his death. And he did it himself, he decided so, he got into the loop, he was still so romantic, he says: with the music, he went away… And I would think so, but seriously, I did not throw him in that loop.*
(Asta, 36)

In one case, the priest was the person who noticed that the bereaved required psychological and psychiatric help:


*He noticed me, and he said… you must worry about yourself. There has been where people just from that high tension are getting problems… He apparently saw that already… I am already disappearing because my weight and everything… I was talking there, I did not even see anything, but I already had those somatic problems… And the priest drove me to the Crisis Centre. Nobody would help, only him.*
(Liepa, 49)

It was important that the priest was friendly and understanding. For example, that he would reassure the bereaved that the suicide is no longer condemned in the Catholic church, at least when it happens in a not wholly conscious and healthy state:


*This bishop goes… And he talks to parishioners… I say I do not know how to approach you, but my husband hanged himself… So, he will never get forgiveness?… He took my hand, held it, and said, we are all invited to this earth… No religion appreciates suicide… it can be a condemnation for seven generations, like a cross, a sin… However, if at that moment when you do this… I even cannot say the word ‘suicide’… When you do this, some button is on, I believe, that at that moment there is like a trance, darkness, you are not able to think anything else.*
(Dalia, 62)

Several participants spoke of other people who helped in rituals or spiritual ways after the loss. Emotional as well as financial support during the funeral were perceived as very important:


*I do not know what I would do if not for my friends. Four friends of him came to the funeral and three hundred friends of mine, who supported me… I do not remember, but there were rows. Moreover, they gathered for me 10,000 euros during the funeral.*
(Asta, 36)

One participant met a Muslim friend, who helped to cope and was, from her point of view, less critical than Catholic people:


*I see that he (the Muslim friend) understands that this is terrible and hurtful, but he believes in the afterlife…, and he thinks that everything is in God’s hands, nothing happens without reason, we met not without reason… He never criticized my views, he is interested in how I see the world, and I am interested in how he sees the world.*
(Nida, 35)

In some cases, other people shared dreams as a message from the deceased, showing their involvement in the grief process:


*I met this nurse… believe it or not, she says, “I do not know, to say or not to say”… She says, I dreamt (the deceased husband)… I knew him… I stand there, and he sees me and goes to me, goes goes goes, says, how great is this, that I met you… You can pass over to (the bereaved woman), that the reason, why it happened… is that I killed a person in a car accident, I hid it, and I could not live with it… The first thought to me, because he hid so much from me… This could have been true.*
(Liepa, 49)

#### 3.2.4. Needing Spirituality-Related Methods to Continue the Bond with the Deceased

For six participants, spirituality-related methods were also crucial to managing the continuing bond with the deceased. For some, the ritual of visiting the grave was a method to remind oneself of the fact of death:


*I need some kind of process, some kind of ritual, which would help me understand and accept. So, I drove to his grave. Traditionally, I lit a candle.*
(Eglė, 31)

For others, rituals helped to connect to the deceased in different ways. For example, to ease the deceased’s soul’s way to heaven:


*When I am in a city, I always go to the church, light a candle, because the candle is somehow cleansing… lifting his soul… it was always important to me that it would be easier for him, I lit the candles… so he would travel somewhere through the more straightforward path than it was for him here.*
(Laima, 28)

Adjusting the funeral to the personality of the deceased seemed for many participants an essential part of the farewell ritual, potentially to maintain aspects of the dead visibly:


*And his grave… I did not want a common grave. He was a wild person, and I made the grave like this. I planted perennial flowers, which can grow by themselves how they want.*
(Agnė, 36)

The funeral was also a place to talk with the deceased:


*During the funeral I said to everyone, go out and give me half an hour to be with him alone… I lay near him, on the coffin and talked to him… Why did you do that, and for what?*
(Asta, 36)

The other spirituality-related way to communicate with the deceased was, in one case, active praying for him and seeing him as an angel:


*At first, the conversations with him were casual… Now they are… when it is hard for me, for example, I talk to him, ask for help… I feel that he sees me all the time, my good and bad behaviors, he does not judge me, but he understands me… he is my angel, and when I pray, I always pray for him that he would care for my child and me.*
(Laima, 28)

In brief, spirituality provides tools which can help during grief. These tools vary from similar to personal and religious rituals to holding onto some spiritual or religious attitudes. The tools help the individuals cope, as well as to continue the bond with the deceased. Priests, psychologists, and friends helped the participants provide these spiritual tools to manage and discuss helpful attitudes.

### 3.3. Spirituality Makes the Grief Process More Difficult

The role of spirituality during suicide bereavement is not only positive and helpful. Spirituality, for nine participants, made the grieving process more difficult in different ways.

#### 3.3.1. Some Spiritual-Related Attitudes and Rituals Make the Grief Process More Difficult

For eight participants, some spirituality-related attitudes and traditions were experienced as disturbing by the bereaved. This was closely related to the reactions and expectations from other people and internalized attitudes and potential projections on other people. For example, the attitude that suicide is a mortal sin, which meant the condemnation of the deceased, caused fear, and prevented individuals from talking to the priest about other spiritual matters:


*I do not know if I could talk to the priest, it would be hard for me… Although, it would be interesting to find out his views towards people who died by suicide… what do they think, is it a big sin… My father said that for a priest, it is a sin not to live, to go away voluntarily… That is why I was afraid that he would talk badly about my fiancé… I did not want that.*
(Laima, 28)

Several participants experienced prohibition from priests and religious people to cry during their grieving process:


*Strangely, everyone would say, “do not cry because you are holding him here. You are not allowing him to go away… Nevertheless, I still cry. It’s cleansing for me… It eases me, but I am holding him here because of my tears… If you cry, his soul is drowning.*
(Laima, 28)

The religious idea of suicide being a test or a punishment angered the participants:


*There were many opinions from the women, who are standardly religious… That God sends for people that much that they can carry… Yeah… I say, ok, what else can I carry? (Laughing ironically).*
(Elena, 34)

The attitude that it required spiritual strongness to withstand suicide provoked tension, and feelings of being not understood:


*I heard from everybody on my first birthday without my husband… Many of them said that “not everybody is that strong” and congratulated me that they wanted to be as strong as me… I thought about it later, how people see the strength… that you don’t cry, smile, care for the children and move further… But a lot can happen. You just find the space for your emotions… It doesn’t mean that you don’t have them.*
(Elena, 34)

Participants noticed that some people who pretended to want to help after the loss concentrated mainly on spiritual issues and did not notice the psychological pain of the bereaved:


*There were many attacks… for example, somebody brought the book of angels… I sat through this session and thought, what a moment… She (the spiritual leader) told me to do a break, pause, and use something… And then she went, and I sent here the sign of a pause in music… My healthy mind turned on… And I sent it (laughing) and didn’t talk to her anymore… It’s strange that she would want to include me into something and then powder everything, that I am not hurting, because… I AM HURTING… Why should I deny it?*
(Liepa, 49)

Several participants perceived some religious rituals and traditions as focusing too much on death:


*When I started to walk a spiritual path, I understood that any connection to the deceased is idolatry and witchcraft. I don’t understand our festivities, vėlinės (a traditional Lithuanian celebration for the souls of the dead). I don’t go to his grave, don’t light the candles.*
(Agnė, 36)

For some, a loved one’s funeral provoked disgust even years after the loss:


*You know how the season has its perfume. Autumn, the scent of autumn and flowers, lilies. It seemed to me that the death happened again somewhere… My new husband gave me flowers as a gift and those lilies, I went inside my home and… the smell of death, the associations… death and corps, disgusting, even candles, you know, when they stand around the body, and the heat causes the smell of the dead body.*
(Laima, 28)

The funeral for several participants was a problematic, even torturing experience:


*That sitting and watching the dead body is torturing for me, I don’t know, it is like, let’s torture ourselves now and watch the deceased person. To say goodbye is ok, but you need 15 min for that.*
(Liepa, 49)

Some experienced pressure to keep in line with the trends of funerals and organizing funerals because of other people:


*I am determined to follow the trends (of the funeral) because it’s unnecessary for me. It is meaningless to me. On another side, we did it all for the parents. Because they are from an older generation, it was, wow, how important for them.*
(Jurga, 49)

or the pressure to keep in line with cherishing the grave:


*And all those stories about graves and tombstones, and this comparison with other monuments, which are more interesting… Maybe if it would… not be Catholicism, but Protestantism… maybe there would be fewer problems, they like simplicity.*
(Nida, 35)

Overall, funerals caused different problems for many participants. This included financial and organizational issues:


*It was a problem for me where to leave my child with disabilities because nobody knew how to feed him… Children should not be near the coffin… But not everybody has people who could watch them.*
(Asta, 36)

It also included problems related to interactions with other people, who sometimes behaved aggressively and unsupportively:


*Nobody listened to his wish to be cremated because the family was furious and blamed me… The funeral was tragic… It was very hard for me, and his relatives didn’t allow me to his coffin, said terrible things to me…*
(Eglė, 31)

Some participants met insensitive priests, which caused disappointment and anger:


*The unpreparedness of the priest, inability to sympathize, comfort, comment about stuff that we all go through the same stages of life, everybody dies, and it’s a challenge… During the funeral or a month after…, he chooses the text from the Bible about how Jesus heals sick people and raises the dead. He accentuates the text several times, especially the point about blindness. But he should know that the blindness in the Bible is not actual blindness… So, what he wanted to say about my husband was that he had this physical disability, but it doesn’t mean that he was spiritually blind… You must search for metaphors in the Bible and not only cite the text and talk about it that the community member had a visual disability. I didn’t understand what he wanted to say.*
(Nida, 35)

The personal rituals or wanting to adjust the funeral to one’s needs, in some cases, were met with criticism:


*You light a candle–they shout at you: “why did you light that candle?”. But it’s nice for me, it helps me. I wanted to light the candle at this place… It’s my spirituality and faith.*
(Eglė, 31)

#### 3.3.2. Spirituality Related Questions Provoke Confusion after the Loss

Eight participants spoke of spirituality related questions after the loss provoking confusion and doubts. Some contemplated existential or spiritual questions. For example, what do I have to live for?


*It’s sad–you don’t want to eat, to get up… I don’t want to do anything, I don’t have what to give to others, and my energy level is too low. I am on a line of exhaustion. And then the questions came: if I don’t have what to give and no energy to live, is it worth it to torment myself and live…? It is the most challenging moment.*
(Eglė, 31)

Some questioned why this had happened to them:


*Why, what, why for me, not for someone else? I didn’t do anything wrong in life. And you start to reconsider your whole life, how, how, how. Maybe you did something to somebody. And of course, people come who “help” you find it.*
(Asta, 36)

The haunting question of responsibility and guilt about suicide was contemplated by each participant. For example:


*And I was sorry before that was guilt because I was angry about many things, mad about him when he was still alive, that these letters from the bank, his debts… I am still mad… And others ask, why did it happen? And I say to them, just because, I don’t know why.*
(Liepa, 49)

The experience of searching for spirituality after loss was perceived as dynamic and emotional:


*You know how you go from one extreme to another… you start to blame God, that he is terrible and how could he let it… you start to pray… then start to hate everybody, God doesn’t exist, every human is awful… Then you begin to love everybody. Then you start to hate everybody because you need help but don’t know what helps, and you are mad at people who can’t help you.*
(Asta, 36)

#### 3.3.3. A Loved One’s Suicide Causes a Spiritual Crisis

For five participants, a loved one’s suicide caused a spiritual crisis. Participants described the spiritual state after the loss as cancer of the soul:


*This cancer of the soul eats you from inside. It spiritually kills you… You don’t want anything; you don’t believe in anything… I called myself a mass then. The mass, which doesn’t think, is languid. And to gather this mass inside, you need a lot of effort… A lot of self-reflection.*
(Asta, 36)

In some cases, the spiritual crisis was primarily related to seeing suicide as treason or abandonment of the bereaved; therefore, their trust in people and love was shattered:


*And you know, we agreed nicely to each other, cherished one another, and then you feel abandoned, betrayed… that he left me.*
(Asta, 36)

Suicide seemed, for some of the participants, as a breaking of spiritual rules:


*I would not want that my daughter would know what suicide is. That there would not be such a concept–to kill. Who tells the people that they can kill themselves?… I disagree with that.*
(Eglė, 31)

A couple of participants said that they were not spiritually prepared for such a disaster; therefore, the effects of the loss were strong:


*There was one more sign before… I wanted to volunteer at a suicide helpline… I wanted to go to the courses… I went through. And in the end, there is like an exam… And I didn’t pass this exam… Later I understood that I would not be able to withstand this topic. All the depression, children. These hurtful topics, only during the courses do I comprehend that it is seriously hard… Later I thought that… the life led me to this helpline, that I would know more about depression, but I gave up. I didn’t understand that I needed this.*
(Jurga, 49)

Participants spoke of disappointment in previous beliefs. For example, disappointment in the Catholic community:


*In the church, there is a mass paid for. And the priest doesn’t hesitate to take the money for the mention of the deceased…, and he also decides to congratulate a birthday person. On the same mass… And when you stand in the first row, there is a candle in front of you… And the choir sings “happy birthday” I just want to take the candle holder and throw it into that priest… Because it is a complete disrespect to people, their grief… I started not to believe in religion, especially in Christian people.*
(Nida, 35);

in good God and His helpfulness:


*There is this anger if God exists, why is he doing that. Why do others live without any losses? Everything is good for them… And for me, punch after punch… When will it end? Or why? I say to God, why do you do that? Do you want to kill me? I don’t have the strength anymore.*
(Asta, 36);

in existential psychology, which previously was important and fulfilled spiritual needs:


*We soaked in existentialism together (with the deceased husband). Heidegger and all other saints, I used to say to him… And I started to think that it was too narrow for me… I said to him, there are many other thinkers, other thoughts, and perspectives on life, not only existentialism… It is too narrow… It can’t save me.*
(Agnė, 36)

#### 3.3.4. Comprehending the Finality of Death Requires Significant Effort

Seven participants revealed difficulties comprehending or contemplating the finality of death. Some participants spoke of their doubts and wished to believe in the afterlife:


*I don’t have the answer about the afterlife. I would like to believe that it exists, but I have doubts about it.*
(Rasa, 41)

Several participants could not stop thinking or dreaming about the look of the decaying body of the deceased:


*I dreamt of the decaying body… I think about it, how that body looks now under this clayey earth?*
(Nida, 35)

Participants searched for ways to comprehend the fact of the death:


*There were moments when you spend time with children and catch yourself thinking, how nice it would be if he would be here… If I start to notice myself in these illusions, dreams, where is the person… I would go to the grave. To put it into my head, this is the reality, not what you are thinking about.*
(Elena, 34)

Some tried to hold onto the attitude that death is the end of the relationship with the deceased and that life is for the living:


*The bond is through memories… What we did, how we communicated, talked, but I don’t believe that he… watches me from Heaven… I don’t know his state if he is sleeping, how the Jews say, Sheol, the afterlife, where they wait for resurrection… I don’t raise these questions for myself. I live my life with the living.*
(Agnė, 36)

Others had to actively block the continuing bond with the deceased:


*When he appears in dreams, it unnerves me… We were harmonized when we lived together. I just would comply, not listen… But now I see that two years went by… And I don’t want to comply with him anymore, even in the dream. It’s better that I would take him and push him aside… I don’t need it. I will manage by myself… If you went away, live your life on your own.*
(Jurga, 49)

Some participants expressed beliefs that holding onto the deceased is harmful:


*If you are nearby… you will constantly be with various emotions… His mother goes to his grave every day. I think she harms herself. She takes her energy away from herself… Why fall together?*
(Eglė, 31)

#### 3.3.5. Discomforting Spiritual Experiences

Five participants spoke of experiencing discomforting feelings regarding the deceased, activated through other people, spaces, things, visuals, memories, and sounds:


*It was an all-surrounding, the death… I tried to explain it to my doctor, the paralyzing fear of dark powers, which is very terrifying… It seemed that I was going out of my mind, nobody could understand me, and I couldn’t explain what I feared. I saw something when I walked the dog… I worked at the time and before work at 7.30 a.m. In the morning, I had to walk my dog before work in winter. It was cruel, I went only on the streets, near the cars, but if I looked at the forest, it seemed that somebody was in the woods. I saw the hanging men in the woods.*
(Laima, 28)

Some participants experienced the sense of the presence of the deceased as discomforting and as provoking fear:


*One thing was tricky… the earth was frozen… I cremated him… And they say, we can’t dig, to bury the urn. So, it stood at my home for a couple days… It preyed on me cruelly. And my parents, grandma… they felt that he is nearby… They are not believers in ghosts or else… But they felt that he had walked in the house, only in the corridor, not the rooms… It was a hard feeling inside that he was not away. I didn’t comprehend that feeling.*
(Asta, 36)

Some participants needed to send the deceased away actively and interpreted this sense of presence as related to suicide’s unnaturally:


*And I dream about him, and he is angry. He is gloomy, he is a grim person, and he comes angry in my dreams. I scream at him, go away, and don’t disturb my life anymore… Several weeks ago. He annoyed me, slipping into my nights and days. And I say in my thoughts, go away… Help us, don’t annoy us. Why do you need to come here and be grim, regulate something… Give us peace. And then the spirit goes away… When the person dies his own death, it is not the same. When he raises his hand against himself, it is cruel. The soul flies around, I don’t know, for hundred years.*
(Jurga, 49)

Overall, spirituality can make the grief process difficult because of interactions with other people who express unhelpful and stigmatizing attitudes related to suicide and grief. The rituals become disturbing if they do not meet the personal needs of the bereaved. Suicide can provoke confusing existential and unanswerable questions that must be explored and contemplated. Suicide can also cause a spiritual crisis and lead a person to reconsider their beliefs. Lastly, loss due to suicide can provoke scary and disturbing spiritual experiences, including the feeling that the death was felt nearby.

## 4. Discussion

This study aimed to explore the role of spirituality during suicide bereavement. The findings implicate that spirituality’s role during grief depends on spirituality’s subjective experiences and beliefs. Spirituality can become a supportive resource that helps cope with various challenges after the loved one’s suicide. It can be perceived as coming without conscious involvement, potentially coming from Higher powers or something more significant than a human being. The bereaved person can also reach for a connection to spirituality by making a specific effort, which also helps during suicide bereavement. Spirituality can provide helpful ways to cope with grief and manifest as active actions, such as rituals, or as beliefs and attitudes a person holds onto. Spirituality also provides methods to form a continuing bond with the deceased and inspires interactions with other spiritual or religious people or discussing spiritual matters even with not necessarily religious people. Lastly, spirituality can make the grief process more difficult through discordant traditions and rituals, interactions with unsupportive religious people, experiences of spiritual crisis or frustration about existential questions. We discuss the main findings below.

### 4.1. Spirituality Is a Supportive Resource That Can Be Reached for or Come without Conscious Involvement

The strengthened spirituality, intuition, a feeling of support from a Higher power and experiencing a continuing bond with the deceased helped the participants cope with complicated feelings of grief. Studies reveal that higher religiosity and positive spiritual/religious coping are related to better bereavement outcomes [28,29,56,57]. The sense of presence and continuing the bond with the deceased as an outer figure can be helpful during bereavement to assure the existence of the afterlife with a hope to meet again and not lose the connection to the loved one [6,8,23]. Our study revealed an additional aspect of spiritual resources during bereavement being perceived as activated without one’s wish. From an analytical psychology point of view, we interpreted this finding as an example of the psyche’s tendency to self-regulate even in the most challenging situations, such as grief [20]. The more the psychological damage, the more definite an answer is given by the unconscious mechanisms to heal the psyche.

In some cases, the bereaved person feels the need and ability to actively invite spiritual resources into the grief process through praying, spending time in sacral places or listening to music. Conscious involvement might be necessary as an active connection with a sense of control and may be a symbolic invitation to start a dialogue with the conscious parts of the psyche [58]. According to the religious coping model, a person can use behaviors, cognitions, emotions, relationships, and virtues related to the sacred domain to cope with complex life events [31,32]. Our study findings show that a person can consciously choose to lean on positive coping strategies related to religion or spiritual domains and find comfort.

Spiritual growth, together with strengthened compassion and responsibility for other people after the suicide loss can be seen as a manifestation of post-traumatic growth [33,59,60], which aligns with the previous findings [6]. Positive religious and spiritual coping can induce post-traumatic growth in different life situations [31,32,61]. In the Jungian approach, encounters with the reality of death can be seen as an unwilling encounter with the personal and collective shadow [20], defined as an unconscious part of the psyche which accommodates everything and that the conscious ego refuses to acknowledge [62]. Since death is culturally and personally repressed [63], facing such a loss, mainly because of an even more socially repressed reason such as suicide [6,24], causes the disturbing experience of meeting one’s own and even a collective shadow. Although renewing the conscious connection with the shadow is complicated and challenging, the positive outcome can manifest as being brave and open to other people’s pain and shadowy experiences. The loved one’s suicide can be seen as a provocation of the whole established system of a person’s conscious–unconscious connection. This can lead to dangerous consequences. Therefore, the concept of post-traumatic growth should not be seen without acknowledging the distress of the loss [64]. However, in some cases, the provocation can translate into personal growth.

Beliefs in a Higher power helped the bereaved with meaning-making of the loss. The theory that grief can be processed through meaning-making in the practical, personal, existential, and spiritual sense is related to the idea that the loss shatters the whole person and their belief system [65]. Religious coping is seen as a helpful way to make meaning of difficult life events [31,32]. Our study revealed that different spiritual and religious beliefs provide ways to make meaning and find some relief during suicide bereavement. Religious and spiritual perspectives offer many insights and traditions about grief, death, the afterlife and how to cope [56,66], which helps the bereaved explain the tragedy and find some sense of meaning in it. Suicide usually leaves the bereaved person with a haunting question of “why” [26,27]. Therefore, if a person believes in religious or spiritual explanations, it can be helpful [6,8,34]. The meaning-making of suicide through explaining its reasons and God’s involvement in the process seem to structure a chaotic and emotional experience of grief after suicide. This could be made from the conscious position but should be congruent with one’s inner beliefs. Our study revealed that constantly reviewing this congruency and adjusting the ideas and explanations is a part of coping with grief.

### 4.2. Spirituality Provides Helpful Ways to Cope with Grief

Our findings revealed and expanded the understanding of the role of spiritual activities, such as rituals, during suicide bereavement. Personally meaningful rituals were the most helpful for the participants to live day by day after the loss. This is a conscious choice to take a specific action [8] which has a personal meaning for the bereaved. The rituals vary from connecting to the deceased or relating to God, lighting candles to calming oneself down before sleep with gratitude, meditation, yoga, or an analysis of the day. Rituals can be helpful because they assure a person of not being condemned by society, as certain rituals, such as funerals, are performed with others [6].

Nevertheless, the effect of the ritual might be related to how much control or personal involvement a person has in creating, organizing or performing the ritual [67]. It could be most important when grief reactions are stronger [68] or, we argue, when grief reactions are provoked by traumatic losses such as suicide. Personal religious or secularized rituals allow for the relief of difficult emotions [69]. They are experiential, culturally ancient [70] methods by which to regain control and act constructively. Rituals as spiritual practices are embodied ways to relate to the deceased, other people, or Higher powers [71]. They can be interpreted as a tribute to the unconscious for more fluent communication [58]. Since suicide loss and grief provoke the experiences of destructive powers of the unconscious [20], rituals as a tribute might function as a regulator.

Symbolic actions can be supplied by spirituality-related attitudes, which help during grief. This includes beliefs in the afterlife, acceptance of uncontrollability and the unknown or trusting God’s will. Holding onto spiritual beliefs and values can be a positive coping strategy during bereavement [28,29,56]. The impact of facing death and experiencing challenging feelings of grief after the suicide of a loved one seem to require more than rational thoughts and understanding of feelings. Searching and holding onto one’s belief system might be helpful after losing a loved one.

Other people, such as those who surround the bereaved, play an essential role in influencing the bereaved’s spirituality after the loss. This is congruent with the general research on religious social support (a kind of support which comes from the religious community) being related to better adjustment to bereavement if the religious support is positive [56] and that bereaved people tend to search for comfort in religious activities and communities [72]. Connecting to other people and receiving spiritual support is another spiritual method by which to cope. The spiritual explanations of suicide and grief from spiritual leaders or spiritual people can function as support. Being together with the bereaved during the funeral can help the person to feel less lonely. The helpfulness of religious communities during grief was revealed in other studies [6,7,25]. Our study demonstrated that the importance of spiritual help can be actualized even if the bereaved is not religious and does not belong to a religious community. In hard times, other people can function as carriers of a part of the bereaved’s feelings, questions, and fears. However, for this interaction to happen, other people must be ready to take on the burden [73].

### 4.3. Spirituality Makes the Grief Process More Difficult

In line with other studies [6,7], our findings reveal the ‘other’ side of spirituality’s role during suicide bereavement. Spirituality-related attitudes, rituals, traditions, questions, and experiences can burden the bereaved. The stigma associated with attitudes toward suicide reach the bereaved through the destructive behaviors of religious people and priests. Stigma might be supported by different religions’ harmful views of suicide as a mortal sin [74]. The funerals and other farewell traditions are not comfortable for everybody but are usually pushed on the suffering. This could be explained by our research participants’ cultural and religious contexts. Suicide, from a Catholic point of view, was looked at as a mortal sin for hundreds of years [75]. Specifically, Lithuania’s Catholicism can be seen as very conservative. During the Soviet occupation, which lasted from 1945 to 1990, the church did not have opportunities to connect freely with the Vatican and develop together but worked as resistance to the Soviet occupation [76]. There are speculations that the church became more political than spiritual [77], which sometimes leaves the religious community as only providing religious beliefs and tools, but with little empathic human contact.

Another aspect of understanding what happens in these problematic interactions is related to caring about other people’s opinions and being unable to confront them. The loss due to suicide and grief is an exhausting, emotionally draining experience [26,27]. Therefore, the bereaved person might not have enough inner resources to express and fight for their opinions and needs. Thus, the need for proactive help to access at least one friendly face after the loss [27] is crucial.

Existential questions, difficulties in comprehending the death, discomforting spiritual experiences, and spiritual crisis affect the bereavement process. On one side, it makes the bereavement process more difficult; on the other side, it can also be seen as a naturally occurring task, which must be in some ways resolved after the loss to make meaning [65], to reach the integration of the loss [27] and to be able to live further. The loss due to suicide can shatter the whole belief system of the bereaved, which causes distress, hopelessness, and distrust in Higher powers [6,7]. From an analytical point of view, the constant inner contemplation of the spiritual experiences, questions, and reconsideration of one’s beliefs might be a component of the grief process [20], which requires time, effort, support, and sometimes help from spiritual leaders and clinicians.

### 4.4. Practical Implications

Our study provided critical practical implications. Practitioners who work with people bereaved by suicide should consider the role of spirituality during suicide bereavement. If necessary for the bereaved, spirituality should be viewed as an additional resource after the loss [6,25]. Acknowledging the importance and discussing rituals, spiritual methods, and helpful beliefs can help with meaning-making after the loss and provide ways to reach emotional relief. Informing the bereaved about the possibility of meeting with a spiritual leader to talk about spiritual crisis and existential questions might also be a productive way to activate the spiritual resources during grief. The practitioners could also receive specific training on how to deal with spiritual issues. They should also be aware of strong emotional reactions to spiritual experiences, help understand these feelings and find constructive expressions for those emotions.

Spiritual leaders should reach for better preparedness and knowledge about bereavement after suicide. Their support and help in spiritual and emotional contexts are required by people who are bereaved by suicide [6,7,25]. Spiritual leaders could normalize the spiritual experiences and issues the bereaved is confronted with. The spiritual explanations for death, suicide, and grief can provide the bereaved with new perspectives about loss and bereavement. Patiently accepting that the suffering might not hold onto specific beliefs and that beliefs and virtues can change during grief is essential. Lastly, spiritual leaders should be responsible for educating their communities about healthy attitudes toward suicide and inspiring compassion in the spiritual members. Otherwise, insensitive comments to the bereaved by suicide or the ignorance toward the psychological state of the bereaved can hurt and challenge the bereaved during the already uncertain time after a loved one’s suicide.

Together with other authors [78], we emphasize the importance of spirituality in postvention strategies, which should involve spiritual support as a potential helping strategy for people bereaved by suicide.

### 4.5. Limitations and Future Research

The current study suffers from several limitations. With its purposive sampling, the explorative qualitative study does not allow for generalizing the findings. The women who participated in the study were volunteers who wanted to tell their stories, which could mean that they had more positive experiences of the spirituality manifestations during grief than those women who did not want to participate. Although homogenous, the sample was only composed of women who lost their life partners, which allowed us to focus on the phenomenon of spirituality but did not allow us to make conclusions about spirituality during suicide bereavement in other groups of bereaved people. The relationship with the deceased differed by the duration and status of the relationship, which might have affected the loss differently. The participants also discussed spiritual and religious beliefs, which provided a broader view of spirituality although this may have made the results more wide-ranging but not as in-depth. Since spirituality might change during a lifetime, the differences in participants’ age might have also biased the study results.

Despite these limitations, our study contributes to the existing literature on spirituality during suicide bereavement. It provides exceptional insights into spirituality as a resource, as a method and as a disturbance during the process of grief after suicide. These findings also raise questions for further research.

Future studies should explore the role of spirituality during suicide bereavement in broader groups of bereaved people due to suicide. They could also expand the research topic to comparisons between the groups based on the person lost due to suicide or based on gender with a deeper exploration of the differences between gender, sex and spirituality during suicide bereavement. For the sake of postvention, research should also focus on spiritual help, spiritual leaders’ preparedness to support the bereaved and the manifestations and integrations of spiritual service in psychological or psychotherapeutic use. With a quantitative approach, future studies should investigate the risk and protective factors related to spirituality during suicide bereavement; more importantly, the positive/negative religious coping strategies and suicide bereavement outcomes should be explored to expand our findings on the different forms that spirituality that can take during suicide bereavement.

## 5. Conclusions

The current study explored the role of spirituality during suicide bereavement and found that the role varies from helpful and providing methods to cope to disturbing and making the grief process more difficult. Findings reveal the helpfulness of personal adjusted rituals, practices, beliefs, and qualitative relationships with spiritual leaders and other people during grief. However, stigmatizing religious attitudes and not personally meaningful traditions provoke distress, anger, and disappointment. Difficult spiritual experiences and spiritual crises might be an intricate part of the bereavement process. Practitioners, spiritual leaders and postvention creators are invited to address the role of spirituality while working with people bereaved by suicide.

## Figures and Tables

**Figure 1 ijerph-19-08740-f001:**
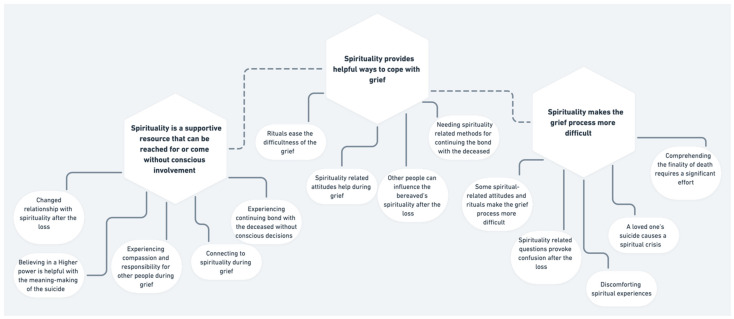
The thematic map.

**Table 1 ijerph-19-08740-t001:** Characteristics of the interviewees and interviews.

Nickname	Age	Deceased Was Her	The Method of Suicide	Age of the Deceased	Time Elapsed After Suicide	Years Spent Together with the Deceased	The Number of Children	Education	Religious Affiliation	Place of Living	How Was the Interview Conducted?	Interview Duration	How Was the Participant Reached?
Rasa	41	Husband	Hanging	44	2.2	20	3	Higher	Catholic	Rural area	Videocall	1:49	Filled the questionnaire
Nida	35	Husband	Overdose and hanging	32	4.9	8	0	Higher	Not affiliated	City	Videocall	1:58	Filled the questionnaire
Liepa	49	Husband	Hanging	44	5	26	2	Professional	Catholic with a question mark	Town	Videocall	1:36	Filled the questionnaire
Laima	28	Fiancée	Hanging	30	4.5	6	0	Higher	Catholic	City	In person	1:58	Filled the questionnaire
Karolina	46	Husband	Hanging	55	2	26	2	Higher	Catholic	City	Videocall	1:01	Proactively called by researchers
Eglė	31	Romantic partner	Hanging	34	2.4	1.5	0	Professional	No affiliation	City	In person	1:43	Proactively called by researchers
Elena	34	Husband	Hanging	32	2.4	17	2	Higher	No affiliation or all religions	City	In person	1:13	Proactively called by researchers
Jurga	49	Husband	Hanging	47	2.1	26	2	Higher	Not practicing catholic	City	In person	1:35	Filled the questionnaire
Agnė	36	Husband	Cutting veins	44	5	8	2	Higher	Catholic	Town	Videocall	1:28	Filled the questionnaire
Dalia	62	Husband	Hanging	57	5	39	2	Higher	Catholic	City	Videocall	2:03	Proactively called by researchers
Asta	36	Husband	Hanging	32	4	17	1	Higher	Eastern orthodox	City	Videocall	2:09	Proactively called by researchers
Average	40.64			41	3.59	17.89						1:41	
SD	9.98			9.63	1.35	11.2							

## Data Availability

Not applicable.

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
