# Peer review of "The Role of Spirituality during Suicide Bereavement: A Qualitative Study"

_ijerph, 2022, doi:10.3390/ijerph19148740_

Round 1

Reviewer 1 Report

This study represents an important qualitative approach to the study of the role of spirituality in suicide bereavement. Congratulations to the authors for their work and for shedding light on a little-studied problem. Suicide rates are increasing and the results of the work may assist in the development of holistic care plans for people bereaved by suicide that include the spiritual needs of these individuals where they exist.

INTRODUCTION

The introduction is a great synthesis of the problems under study and adequately justifies the need to carry out studies of this type. I would also like to make a number of suggestions that I believe could improve the quality of the introduction.

When showing the incidence of suicide, it may be useful to mention data on suicide within the context in which the study is conducted. In some European countries, the suicide rate is above the world average. If this is so in the case of Lithuania, it may be a further element of justification for the study, and puts the reader in context.

Spirituality and religiosity are sometimes used interchangeably, or at least spiritual people are understood to be religious (spirituality being defined as the search for the sacred). It is adequate, but I think it should be justified. Perhaps one solution is to define spirituality more broadly, and also to define the term religiosity. In addition, it may be useful for the reader to be aware of the existence of other types of spirituality that do not involve religiosity. Here are some articles that may help in this distinction:

https://doi.org/10.1016/j.jesp.2016.11.006

https://doi.org/10.1163/157361212X644495

https://doi.org/10.1097/NMD.0b013e31816ff796

Religiosity is defined as a method of searching for the sacred. Therefore, it is difficult to distinguish it from the definition given for spirituality. It also implies that there are other methods of searching for the sacred that have not been taken into account. Returning to the above, I believe that to alleviate this problem it is best to define both terms broadly and try to distinguish between them without losing sight of the fact that, for the purpose of the paper, they are related terms.

Given that most of the sample is defined as Catholic (practicing or not), I suggest that the authors mention the particularities of suicide bereavement coping in Catholic people. I recommend you discuss the differences between religions and suicide. This other article might help: DOI: 10.1007/s10943-020-01137-x

As a final suggestion, since the potential of spirituality to cope with bereavement is sometimes mentioned, it may be appropriate to mention Pargament et al. (1997; 1998), who developed the concept of religious coping in stressful situations. (https://doi.org/10.2307/1388152)

MATERIALS AND METHOD

The information on materials and methods is extensively described and adequately justified.

In the subsection on participants, it might be appropriate to mention the type of sampling used (in this case, it appears to be a non-probability snowball sampling in some cases, and purposive in others).

RESULTS

The results are very complete. Its correct division by subject matter makes it easy to understand. Despite its breadth, it turns out to be a great exercise in synthesis.

DISCUSSION

The discussion is adequate, relates the findings to previous findings and thus allows understanding the implications of the study. However, I would like to make a few comments.

On several occasions, data are interpreted from the perspective of analytical psychology (e.g., lines 803 and 906). This perspective of psychology was already mentioned in the introduction (line 34). Why are the results interpreted from this perspective? Although not incorrect, perhaps it should be justified. It could probably be done through the legacy of Jung and his study of religion (as already done elsewhere in the discussion) or the idea of repression of traumatic events. However, other perspectives should be mentioned, perhaps with a more social focus, given the implications that both suicide and religion/spirituality have at this level.

Special emphasis is placed on spirituality as a resource. Perhaps here we can open the range and not only use this analytical perspective, but also mention more broadly ideas such as the positive and negative religious coping developed by Pargament et al. in relation to the different ways of coping with bereavement: some that facilitate post-traumatic growth and others that hinder it.

The involvement of God in the bereavement process is also mentioned as something that facilitates the bereavement process. This may depend on the type of relationship the subject has with God. It may indeed be that a positive relationship with God facilitates the coping of such a process. However, there are forms of negative relationship with God, related to negative coping, that can complicate a stressful process such as suicide bereavement. Some constructs in this regard may be mistrust in God or religious/spiritual struggles. It may be useful to review the literature in this regard or to point out the need to expand research on these types of variables in relation to bereavement.

In section 4.2 it may be useful to talk about religious social support, a type of social support derived from the individual's religious participation.

In the implications, I think that in addition to mentioning spiritual leaders and the idea of professionals referring people with spiritual needs for bereavement to these leaders, one could mention the possibility of professionals (such as psychologists) receiving training on how to deal with spiritual issues.

Reviewer 2 Report

The proposal presented by the authors is relevant from a social and academic point of view and is well written. Just a few suggestions in order to, in the opinion of this reviewer, facilitate the reader's understanding and enhance the contribution of the text to the development of knowledge in this area:

Regarding the summary, the Keywords should include “spiritual copying”, and probably some other term would enrich the contextualization of the article once the suggestions indicated below have been considered. Also the summary, in particular the results, could be described in a more complete way, reflecting to a greater extent aspects that appear in the Discussion section. In this way, the summary will also be more attractive to invite those who start from its previous reading to read the full text.

The authors make an adequate introduction that situates the topic in an appropriate way by indicating the social relevance of the topic and placing it in the field of studies on religious and spiritual coping. Although I believe that it is not necessary to further develop the concept of spirituality, I understand that between the penultimate and penultimate paragraphs it would be necessary for the authors to further develop the mechanisms and processes by which religion acts as a coping resource, being able to develop aspects around K. Pargament's model of religious coping. I understand that this is necessary to later understand what the role and  the ways in which spirituality acts as a coping resource are. The authors should also discuss their results in this context, analyzing the similarities and differences between religious and spiritual coping processes.

The penultimate paragraph, which points out some aspects studied from the religious and spiritual spheres around suicide, should also indicate what the results of these studies are, and not limit itself to indicating the constructs on which it has been studied. It would be convenient to highlight the qualitative studies, in order to point out what has been studied and is known from this methodology in previous research in order to highlight and describe the contribution and novelty of this study to the development of knowledge in this area.

Given the special significance the selection of study participants has in this study, this aspect requires further consideration in the first introductory section of the study. The authors must highlight and give value to the gender orientation of the study, explicitly showing the gender differences in the previous studies cited above, or mentioning the absence of these differential studies, if that were the case. The introduction should also include a greater description of the nuances that this distinction based on sex (man-woman) can contribute to the description of the processes and mechanisms of religious and spiritual coping. Finally, the authors should highlight the contribution of their study and approach to the development of knowledge in this specific field.

Regarding the methodology used, this should be described in greater detail, not only defining what reflexive thematic analysis consists of, but also indicating whether it was deductive or inductive and why, and describing somewhat more the conceptual particularities of the thematic analysis approach used in this research. It is important that the authors develop these aspects in a much more detailed way.

In my opinion, the narration of the results should be a bit more structured and precise, and it is possible that the literal examples based on interview transcripts would be better contextualized by the reader if the woman's name was accompanied by age.

The first paragraph of the Results section includes various aspects from lines 175 to 197 that do not go beyond what the authors collect at the beginning as an understanding of the phenomenon of spirituality by the participating women. It would be convenient not to dispense with the information regarding the religious affiliation of the participating women. However this should not be linked to particular expressions of spirituality. As the information is presented, it is difficult to associate both types of information. I would suggest separating both types of information into separate paragraphs. Also given that the focus in the presentation of results is oriented towards thematic analysis, the presentation of these ideas about how the vision of spirituality present in women should also focus on common (and different) themes or ideas, instead of depending on the participating women. The speech, in this sense, would be more adequate to the objectives of the study. Since this is the first part of the results section, and in line 205 the description of the first theme observed begins, it would be convenient to structure the Results section in sub-sections, differentiating this first part of the detailed description of the themes that they are described from the current section 3.1.

The title of section 3.1.1. it's confusing. I understand that the title mentions a change in the person's relationship with spirituality, but what is being pointed out in lines 210 to 261 are spiritual changes perceived by the participating women. In my opinion, the title of this sub-section should be revised to better reflect the content of the section.

It would also be convenient that the first statement, which seems to summarize the contents that are developed later, as well as the title of section 3.1. be reviewed:

3.1. Spirituality is a supportive resource that can be reached for or come without conscious

involvement

“All the participants spoke about spirituality as a supportive resource. The resource 207

sometimes was activated through active involvement, but mainly it functioned without 208

one's conscious efforts."

While the text includes changes that are perceived by the participants, their unconscious dimension is not so evident. This does not appear directly in section 3.1.1, and section 3.1.2., is oriented in another way and towards a different construct, which is the construction of meaning and not changes in spirituality.

It is possible that the use of the term unconscious is not appropriate in this context, since the discourse is indicating the absence of intentionality, decision, or initiative, on the part of the woman interviewed (heading 3.1.5 should also be revised).

The title of section 3.2. it should include the term religion to more accurately reflect the content of the section, which explicitly includes and mentions religious aspects.

Also review the titles of the sub-sections, so that they reflect the relevant content in a more precise and informative way (rituals, attitudes, personal relationships, …)

Section titles should describe the topic, not the conclusion of that section. Examples:

Some spiritual-related attitudes and rituals make the grief process more difficult

Spirituality related questions provoke confusion after the loss

A loved one's suicide causes a spiritual crisis

Comprehending the finality of death requires a significant effort

The Discussion Section:

In my opinion, the way in which the unconscious vs. conscious reference is used to explain the function of spirituality as a coping resource does not correspond to the experiences and situations that, based on what was observed in the sample, the participants report.

Although the authors' point of view, as an unconscious aspect, could be presented in the discussion as a possible interpretation of the results, it does not seem justified to assume this as the first of the explanations if we start from what was reported in the responses of the participants in the study. The discussion approach, based on the analytic theory is adequate and pertinent but could be completed with additional theoretical contributions such as the "new unconscious", satisfaction of needs (spiritual, cognitive, construction of meaning...), phenomenological humanistic psychology, psychological existential perspective, etc. Some of these contributions appear in the discussion but their exposition should be carried out in a more orderly manner.

Likewise, the discussion could contemplate the three orientations that appear in the experiences reported by the participants in the study: interpretations that allow understanding from a theoretical point of view the increase in spirituality/religiosity, the situation of crisis or rejection, or its maintenance . Likewise, it would be positive if the authors also considered at some point in the Discussion section the idea that they present in the introduction regarding the role, generally negative, that suicide plays in the main religions. Section 4.3. indicates some aspects, but the discussion would be richer if it were completed with aspects of a theological nature typical of different religions or spiritual orientations. This section should be completed as indicated.

Section 4.2 of the Discussion is appropriate and well focused, although the last part, where the relational dimension of spirituality is addressed as a coping resource, should be completed with a deeper analysis that investigates the motives, from a psychological point of view, for which social support (spiritual in this case, although it could also be religious in the case of some of the participants with a more institutional experience of their faith), performs each of the positive functions that are cited in the last paragraph of this section (4.2.).

This Discussion section should also be completed with a gender focus. Although it seems appropriate that the fact of using a sample composed only of women be pointed out as a limitation of the study, this does not exclude the fact that the authors take advantage of this opportunity to enrich the existing literature with a gender approach in which the results observed, and the conclusions and discussions of the results, contemplate aspects such as the following:

• Evaluate the results observed in the responses of the participants in the historical moment and the cultural context from which the participants come.

• Also assess these results considering, to a greater extent, social and cultural aspects that can account for the results (this aspect is considered to some extent by the authors in the discussion).

• Analysis of possible differences or inequalities in comparison with previous studies or known aspects in relation to the masculine.

• Place the results also in the context of the age of the study participants (and other relevant sociodemographic characteristics if any).

• Investigate possible situations of inequality associated with the situation described in the study: spiritual coping with grieving situations due to the suicide of a close person.

In addition to this gender perspective, it would be convenient to place the results of the study in the context in which the article will be published, as well as the moment in which the study data is collected, mentioning their relevance, and the possible influence and relationships, with the social and spiritual situation experienced worldwide in relation to the Covid-19 pandemic.

Although the authors suggest that future research should focus on studying postvention to a greater extent, without going into very detailed explanations, the authors should add a few more lines to this paragraph (lines 929-934) pointing out some of the implications from their point of view. , of the results of this study in relation to postvention. Another option would be to complete the last section (4.5) with these aspects.

Finally, point out the need to strengthen some of the statements and suggestions made in this last section with supporting bibliographical references.

I would also suggest anticipating this section 4.5. before the limitations and contributions of the study, and end the Discussion section with a few brief conclusions.

Reviewer 3 Report

The authors present a well-executed and clearly written study on the role of physical, emotional, psychological and social aspects of spirituality and religion in bereavement due to suicide. The study provides nuanced and clear insights into the experiences of women who have lost their partners to suicide and how they interact with spirituality in this experience. If I was to give one point of critique: it does not become fully clear how the sacred is at stake in this research, which is presented as the defining feature of spirituality. 

Author Response

Response to Reviewer 3 Comments

Comments and Suggestions for Authors

The authors present a well-executed and clearly written study on the role of physical, emotional, psychological and social aspects of spirituality and religion in bereavement due to suicide. The study provides nuanced and clear insights into the experiences of women who have lost their partners to suicide and how they interact with spirituality in this experience. If I was to give one point of critique: it does not become fully clear how the sacred is at stake in this research, which is presented as the defining feature of spirituality.

Response 1: Dear reviewer, thank you for your positive comments about our article. We made english language corrections. We also added more about the definition of spirituality in the introduction section and more interpretations (using more models of religious and spiritual coping and support) of the findings in the discussion section, which, we hope, can make clearer the researched phenomenon of spirituality and how it was experienced during suicide bereavement.

Reviewer 4 Report

In the Abstract the keyword “qualitative” is useless.  “Religion” for instance, could say more about the content.

In the Introduction (line 84), please add some sentences explaining the reason why “The suicide bereaved life partners can have even more expressed complicated grief than friends, parents or colleagues and children or siblings”, so the reader will immediately realize the merit of the sample chosen for this research

Author Response

Response to Reviewer 4 Comments

Dear reviewer, thank you very much for your comments on our manuscript. We improved the article according to your comments.

Point 1: In the Abstract the keyword “qualitative” is useless.  “Religion” for instance, could say more about the content.

Response 1: We deleted the keyword “qualitative” and added “religion”. We agree that those keywords are more adequate for the article.

Point 2: In the Introduction (line 84), please add some sentences explaining the reason why “The suicide bereaved life partners can have even more expressed complicated grief than friends, parents or colleagues and children or siblings”, so the reader will immediately realize the merit of the sample chosen for this research

Response 2: We added a sentence about the potential reasons, which we found in the literature. What was interesting, that additionally, we found that there is very little known about the actual reasons, the authors only wrote about their assumptions, but the more research is needed for this, too. Thank you for this comment.